# Asparaginase-Phage P22 Nanoreactors: Toward a Biobetter Development for Acute Lymphoblastic Leukemia Treatment

**DOI:** 10.3390/pharmaceutics13050604

**Published:** 2021-04-22

**Authors:** Cristina Díaz-Barriga, Francisca Villanueva-Flores, Katrin Quester, Andrés Zárate-Romero, Ruben Dario Cadena-Nava, Alejandro Huerta-Saquero

**Affiliations:** Centro de Nanociencias y Nanotecnología, Universidad Nacional Autónoma de México, Km. 107 Carretera Tijuana-Ensenada, Ensenada 22860, Mexico; dbh.cristina@gmail.com (C.D.-B.); villanueva@cnyn.unam.mx (F.V.-F.); quester@cnyn.unam.mx (K.Q.); azarate@cnyn.unam.mx (A.Z.-R.); rcadena@cnyn.unam.mx (R.D.C.-N.)

**Keywords:** asparaginase nanoreactors, ALL treatment, P22 virus-like particles

## Abstract

Asparaginase (ASNase) is a biopharmaceutical for Acute Lymphoblastic Leukemia (ALL) treatment. However, it shows undesirable side effects such as short lifetimes, susceptibility to proteases, and immunogenicity. Here, ASNase encapsidation was genetically directed in bacteriophage P22-based virus-like particles (VLPs) (ASNase-P22 nanoreactors) as a strategy to overcome these challenges. ASNase-P22 was composed of 58.4 ± 7.9% of coat protein and 41.6 ± 8.1% of tetrameric ASNase. Km and Kcat values of ASNase-P22 were 15- and 2-fold higher than those obtained for the free enzyme, respectively. Resulting Kcat/Km value was 2.19 × 10^5^ M^−1^ s^−1^. ASNase-P22 showed an aggregation of 60% of the volume sample when incubated at 37 °C for 12 days. In comparison, commercial asparaginase was completely aggregated under the same conditions. ASNase-P22 was stable for up to 24 h at 37 °C, independent of the presence of human blood serum (HBS) or whether ASNase-P22 nanoreactors were uncoated or PEGylated. Finally, we found that ASNase-P22 caused cytotoxicity in the leukemic cell line MOLT-4 in a concentration dependent manner. To our knowledge, this is the first work where ASNase is encapsulated inside of VLPs, as a promising alternative to fight ALL.

## 1. Introduction

Acute Lymphoblastic Leukemia (ALL) is a hematologic disorder in the bone marrow, blood, and extramedullary sites characterized by the uncontrolled proliferation of immature lymphocytes. Lymphocytes stop developing in an early stage of cell differentiation and accumulate rapidly, thereby displacing normal hematopoietic cells in the bone marrow [1]. ALL accounts for approximately 25% of all childhood cancers and about 80% of all leukemias, representing the most common type of childhood cancer and the leading cause of death from disease in children [2]. ALL affects both children and adults, with a maximum incidence between the ages of 2 and 5 years [3].

The most common drugs used for ALL treatment are vincristine (an antineoplasic alkaloid), corticosteroids, and Asparaginase (ASNase; EC 3.5.1.1) [4,5].

ASNase is found in the three domains of life; it has a great value in the chemotherapeutic treatment of ALL and other types of leukemia such as lymphosarcoma, Hodgkin’s disease, acute myelogenous leukemia, chronic lymphocytic leukemia, reticulosarcoma, and melanosarcoma [6,7].

ASNase is a homotetrameric enzyme that catalyzes the hydrolytic deamidation of L-asparagine (Asn) to aspartic acid (Asp) producing ammonium (NH4) in blood. Leukemic cells require Asn as an essential extracellular nutrient, contrary to normal cells, which are able to synthesize Asn on their own. ASNase exerts a selective toxicity based on the inhibition of cancer cell growth by inducing cell cycle blockade and, finally, starvation-induced apoptosis [8,9]. Currently, four commercial formulations of ASNase for ALL treatment are available: *E. coli* native ASNase II (EcAII), *E. chrysantemi* ASNase (ErAII), pegylated *E. coli* ASNase II (polyethylene glycol covalently linked ASNase; EcAII-PEG), and a pegylated E. chrysantemi ASNase (ErAII-PEG) [10,11,12]. These commercial formulations differ in half-life times and immunogenic effects. Native EcAII is the most prescribed ASNase, with a half-life of 26–30 h. However, it may cause ASNase antibody production and hypersensitivity in 60% of the patients, particularly in teenagers and adults. Resulting reactions can range from mild allergies to anaphylactic shocks and can cause rashes, respiratory problems, low blood pressure, sweating, and loss of consciousness [10,13]. By replacing EcAII with ErAII, immunogenic response can be minimized, but the main disadvantage of ErAII is its low half-life of 16 h. Meanwhile, EcAII-PEG has shown reduced side effects and an elevated half-life of 5.5–7 days and, therefore, a lower dosage is required (2000–2500 IU/m^2^ every 2–4 weeks, compared to 6000 IU/m^2^ thrice a week of native EcAII) [13]. However, this formulation is expensive and difficult to find in developing countries such as Mexico.

New alternatives are needed to improve the pharmacodynamic and pharmacokinetic of ASNase [12,14]. Several strategies have been explored that include structural modification of enzymes, recombinant ASNase production, and enzyme encapsidation, as well as identifying ASNase in other microorganisms [15,16,17,18].

One of the most promising strategies is the ASNase confinement/immobilization in a scaffold/substrate, because it protects the enzyme from the action of proteases and increases the catalytic half-life in vivo [19]. Several organic, inorganic, and hybrid materials have been recently reviewed for ASNase immobilization or encapsidation [20], including lipid Langmuir films [21], liposomes [22,23], erythrocytes [24], and synthetic nanocapsules [25,26,27], among others. However, it is necessary to further explore the encapsidation of ASNase in order to obtain a minimal immunological response upon application to patients.

Virus-like particles (VLPs) are considered as a successful model for nanoreactors and drug delivery systems [28]. VLPs consist of multiple copies of self-assembling proteins, which form symmetric, stable, polyvalent and monodisperse nanostructures that, due to their lack of genetic material, cannot cause infections. Their self-assembly properties allow the formation of a nanoparticle identical to the original virus capsid, but instead of genetic material, it contains a molecule of interest, such as an enzyme or a drug. The application of these systems has been widely studied and represents a revolutionary alternative in medicine [28,29,30,31]. Other advantages of using VLPs as nanobiovehicles are a longer half-life and elevated biocompatibility, and they can also be chemically functionalized and produced in large quantities in a short time [28,32,33]. VLPs of the phage P22 is one of the most studied and applied systems for enzymatic nanoreactors because of its large payload volume and a mechanically robust capsid. P22 VLPs are composed of 420 coat proteins (CP) that assemble with the aid of scaffold proteins (SP) to form icosahedral capsids with a diameter of 60 nm [34]. The in vivo encapsidation of a cargo protein can be genetically manipulated, as shown by O’Neil et al. [35], who generated a P22 VLP with an enzyme as cargo by fusing the gene coding for SP to the gene coding for the enzyme of interest. In another study, Patterson et al. [36] encapsulated an enzymatic cascade composed of three different enzymes in P22 VLPs.

In this work, we report the genetically directed encapsidation of the enzyme ASNase II from *E. coli* into VLPs of the bacteriophage P22 to achieve the enzymatically active ASNase-P22 nanoreactor. Additionally, ASNase-P22 nanoreactors were PEGylated, and their kinetic parameters were fully characterized. Stability at different temperatures and in the presence of human blood serum (HBS) was also studied. Furthermore, the role of glycerol as a stabilizer was investigated. All experiments included the commercial formulation of the enzyme Leunase^®^, used as a control. Finally, ASNase-P22 nanoreactors cytotoxicity was evaluated in the human leukemic cell line MOLT-4, causing cytotoxicity in a concentration-dependent manner. The potential use of ASNase nanoreactors as a novel alternative for ALL treatment is discussed.

## 2. Materials and Methods

### 2.1. ASNase-SP Expression

*E. coli ansB* was amplified by PCR using the Ready MixTM Taq PCR Reaction Mix with MgCl_2_ (Sigma-Aldrich, St. Louis, MO, USA, P4600). During the amplification, the restriction sites NcoI and XhoI were introduced and then cloned into the vector pBAD after double-digestion of both, PCR product and vector. From the vector pBAD *cyp-sp* [37], *sp* was amplified by PCR, conserving a 5′-XhoI restriction site but eliminating a XhoI site within the *sp* sequence and furthermore introducing an EcoRI restriction site at the 3′-end. After double digestion, *sp* was ligated into pBAD *ansB*, resulting in the vector pBAD *ansB-sp*. All the primers used are listed in Table 1. pBAD *ansB-sp* was then transformed into *E. coli* BL21 DE3 pLysS cells according to Pope and Kent [38]. Overexpression of ASNase-SP was induced by 0.125% L-arabinose (Sigma-Aldrich, MO, USA, A3256) and confirmed via SDS-PAGE (12%), as reported by Sánchez-Sánchez et al. [37].

### 2.2. Production of ASNase-P22 Nanoreactors

The pRSF/P22-CP vector, encoding for the bacteriophage P22 coat protein, was transformed by heat shock into *E. coli* BL21 DE3/pLysS that already contained pBAD/*ansB-sp*. Overexpression with 0.125% L-arabinose (Sigma-Aldrich, A3256) (ASNase-SP) and 0.5 mM isopropyl β- d-1-thiogalactopyranoside (IPTG) (Sigma, I6758) (P22-CP) was confirmed by SDS-PAGE (12%).

Differential expression of both plasmids was performed, as described by Tapia-Moreno et al. [39]. Cells were grown in Terrific Broth culture media (TB) with 34 µg/mL chloramphenicol (Sigma-Aldrich, C3175), 30 µg/mL kanamycin (Sigma-Aldrich, 60615), and 200 µg/mL ampicillin (Duchefa Biochemie, Haarlem, The Netherlands, A0104) at 37 °C and 180 rpm until reaching an absorbance of 0.9 at 600 nm. Then, ASNase-SP was expressed by adding 0.125% L-arabinose at 30 °C and 180 rpm for 16 h. Afterward, P22-CP expression was induced by adding 0.5 mM IPTG for 4 h under the same culture conditions. The culture was centrifuged for 15 min at 5000 rpm. The bacterial pellet was kept at −80 °C until use. A schematic representation of the ASNase-P22 nanoreactor production is shown in Figure 1.

### 2.3. Purification of ASNase-P22 Nanoreactors

ASNase-P22 nanoreactors were purified by sonication in lysis buffer pH 7.6 (50 mM NaH_2_PO_4_ (JT Baker, Loughborough, UK, 3818-01), 100 mM NaCl (JT Baker, Loughborough, UK, 3624-01), and 5% glycerol (Sigma-Aldrich, St. Louis, MO, USA, G2025) for 10 min (9.9 s ON, 9.9 s OFF) in a TU-650Y Homogenizer sonicator cell crusher and then underwent ultracentrifugation in 50 mM PBS and 35% sucrose (JT Baker, Loughborough, UK, 4072) for 2 h at 31,000 rpm and 4 °C in a Beckman Coulter, Optima XPN-100 centrifuge. The pellet was resuspended in 2 mL 50 mM PBS pH 7 under constant and slow agitation.

Nanoreactors were then purified using a HiPrepTM 16/60 sephacryl S-500 HR column (Sigma-Aldrich, MO, USA) and an ÄKTA prime plus Fast protein liquid chromatography (FPLC) (GE Healthcare, Waukesha, WI, USA). Fractions containing the ASNase-P22 nanoreactors were stored at 4 °C until further characterization.

### 2.4. Capsid Composition

The percentage of the proteins composing the VLPs were estimated by a densitometry analysis of the SDS-PAGE gels using the program ImageJ (NIH), similarly to the method described by Giessen et al. [40].

The confinement ASNase molarity and the occupation percentage inside the VLPs were calculated according to Equation (1), as previously reported [41].
(1)$$M_{conf}=\frac{\left(enzymes\;in\;capsid\right)\;\times \;\left(\frac{1\;mol}{6.022\;\times \;10^{23}proteins}\right)}{internal\;capsid\;volume\;}$$

The occupation percentage of the enzyme ASNase in the capsid was determined using Equation (2).
(2)$$occupation\;\%=\frac{\left(enzymes\;in\;capsid\right)\;\times \;\left(ASNase-SP\;volume\right)}{internal\;capsid\;volume\;}\;\times \;100$$

Internal capsid volume was calculated, assuming a spherical form by the formula (v = 4/3 πr^^3^) and considering an internal radio of 30 nm, determined by Transmission Electron Microscopy (TEM). Accordingly, the internal capsid volume of the P22 phage was 11.31 × 10^−20^ L.

The volume of the tetrameric ASNase-SP was calculated according to Equation (3).
(3)$$V_{enzyme}=\frac{\left(V_{bar}\right)\;\times \;\left(10^{20}\frac{{\mathrm{nm}}^{3}}{{\mathrm{cm}}^{3}}\right)\times \left(MW\right)}{6.022\;\times \;10^{23}\frac{molecules}{mol}\;}$$
where *V_bar_* is the specific partial volume calculated based on the amino acid sequence of ASNase using the program Sednterp (BITC), obtaining a value of 0.7357 mL/g. *MW* is the molecular weight of the enzyme. The volume of ASNase-SP obtained was 262.19 nm^3^.

### 2.5. PEGylation of ASNase-P22 Nanoreactors

ASNase-P22 nanoreactors were pegylated using PEG-5000 carboxylic acid (Polysciences, Warrington, PA, USA, 26036) in 50 mM PBS pH 7 under the presence of 1-Ethyl-3-(3-dimethylaminopropyl)carbodiimide (EDC) (Merck, Darmstadt, Germany, E6383) and N-hydroxysuccinimide (NHS) (Merck, 130672) at equimolar ratio. Activated PEG5000 carboxylic acid was added to nanoreactors at a molar ratio of 1:100 and incubated under slow agitation for 16 h at 4 °C. To remove the excess of EDC, NHS, and free PEG, the sample was ultrafiltrated through Amicon Ultra membranes with a pore size of 30 kDa (Millipore, MA, USA, PBTK07610).

### 2.6. ASNase-P22 Nanoreactors Characterization

#### 2.6.1. Size and Zeta Potential

The size and zeta potential of the VLPs were determined by dynamic light scattering (DLS) (Zetasizer NanoZS, Malvern, UK) using capillary cells (Malvern Panalytical Inc., Malvern, UK, DTS1070).

#### 2.6.2. TEM Characterization

ASNase-P22 nanoreactors, placed on a copper grid (400 mesh, formvar/carbon, TedPella, CA, USA), were stained with 2% uranyl acetate (Thermo Fisher Scientific, Waltham, MA, USA, NC0788109) and analyzed with a JEOL JEM-2010 TEM (Tokyo, Japan) operated at 200 kV.

#### 2.6.3. ASNase Activity

ASNase activity was determined by the Berthelot’s reaction to measure the ammonia liberated during the deamination of Asn [42]. The method used was that reported by Chaney et al. [43], with the following modifications: 12.5 µL of 6 mM Asn (Sigma-Aldrich, St. Louis, MO, USA, A0884) dissolved in 1 mM PBS pH 7 were added to 2.5 µL of ASNase and incubated at 37 °C. Reaction was stopped by adding 12.5 µL of 1.5 M trichloroacetic acid (Sigma-Aldrich, St. Louis, MO, USA, T6399) at different time points. Then, 2 µL of this reaction were transferred to 96 well plates (Thermo Fisher Scientific, Waltham, MA, USA, 260860) and 18 µL of deionized water were added. Afterward, 100 µL of solution 1 (0.5 M phenol, Merck, Darmstadt, Germany, 242322; 1 mM sodium nitroprusside, Merck PHR1423) were added, followed by 100 µL of solution 2 (0.625 M sodium hydroxide, JT Baker, 0402.0500; 0.03 M sodium hypochlorite, Merck 425044). The reaction mix was incubated at 25 °C for 25 min, and the product reaction was measured in a spectrometer Multiskan FC, Thermo Fisher Scientific, 51,119,000 at 625 nm. A standard curve from 0 to 5 mM of ammonia chloride (NH_4_Cl) (Merck, 254134) was used as a reference.

Kinetics parameters of the ASNase-P22 nanoreactors were determined by plotting the initial rates of ammonia production at different substrate concentrations (0 to 5 mM of Asn). Results were fit to Michaelis-Menten’s model using the software Origin 7.0.

The specific enzymatic activity was determined using the protein assay kit BCA (Pierce, Thermo Fisher Scientific, 23227), using a standard curve of bovine serum albumin (BSA) (Merck, A2153) as a reference. All experiments were compared with free commercial ASNase (Leunase^®^, WI, USA, Sanfer) as a control.

#### 2.6.4. Thermal Stability

Thermal stability was studied by measuring enzymatic activity by incubating the nanoreactors at 4 °C, 25 °C, and 37 °C for at least 48 h, as described above. Enzymatic activity was determined in the presence or absence of 20% *v*/*v* HBS and 10% *v*/*v* of glycerol as a stabilizer. Protein aggregation was estimated by DLS. HBS was obtained by centrifugation of healthy human blood samples (from myself, AHS) for 10 min at 10,000 rpm and collected in a vacutainer blood collection tube (Fisher scientific, 13-680-61).

#### 2.6.5. ASNase-P22 Cytotoxicity

ASNase-P22 nanoreactors toxicity was evaluated against the T-cell lymphoblastic human leukemia cell line MOLT-4 (CRL-1582, ATCC). Cells were plated on 96 well plates (Thermo Fisher Scientific, 260860) at 4 × 10^5^ cells/mL and maintained in the Roswell Park Memorial Institute medium (RPMI 1640) (Merck, R6504), modified to contain 2 mM L-glutamine (Merck, G6392), 10 mM 4-(2-Hydroxyethyl)Piperazine-1-Ethanesulfonic Acid (HEPES) (Sigma, H4034), 1 mM sodium pyruvate (Merck, P5280), 4500 mg/L glucose (Merck, G7021), and 1500 mg/L sodium bicarbonate (Merck, S5761) with 10% of fetal bovine serum (*v*/*v*) (Gibco, Thermo Fisher ScientificS, 11533387). Cells were incubated at 37 °C with 5% CO_2_.

Cell proliferation was measured using the CellTiter 96^®^ AQueous One Solution Cell Proliferation Assay kit (Promega, Madison, WI, USA, G3581), based on the [3-(4,5-dimethylthiazol-2-yl)-5-(3-carboxymethoxyphenyl)-2-(4-sulfophenyl)-2H-tetrazolium] (MTS) assay, following the manufacturer’s instructions at different concentrations of the enzyme. Cell viability was determined after 72 h of incubation.

### 2.7. Statistical Analysis

All measurements reported are expressed as the average ± standard deviation of at least three independent experiments. Statistical significances were calculated by a two-way analysis of variance (ANOVA), followed by a Tukey’s test.

## 3. Results

### 3.1. Expression and Purification of ASNase-P22 Nanoreactors

*E. coli* BL21 DE3/pLysS cells were transformed with the plasmids pBAD/ASNase-SP and pRSFDuet1/P22-CP, and successful overexpression of both proteins, ASNase-SP (53 kDa) and CP (47 kDa), were confirmed by 12% SDS-PAGE (Figure 2).

Bacteriophage P22-based VLPs with ASNase activity were purified by ultracentrifugation and size exclusion chromatography (Figure 2a). Peak 1 represents the biggest particles eluted, where ASNase-P22 nanoreactors were expected to be found. Peak 2 represents other proteins as well as unassembled nanoparticles. Figure 2b shows SDS-PAGE of the purification steps of ASNases-P22 nanoreactors. Lane 1 represents the molecular weight marker. Lane 2 represents non-induced cells lysate used as a negative control. Lane 3 represents induced cell lysate, where we highlighted the two bands corresponding to the molecular weights of ASNase-SP and CP of 53 kDa and 47 kDa, respectively. Lane 4 represents peak 1 of the chromatography, where the bands of purified ASNase and CP are shown as expected. Lane 5 shows peak 2 of the chromatography, with other proteins and unassembled nanoreactors appearing. Red arrows point out the ASNase-SP and CP bands.

Purified VLPs were analyzed by TEM (Figure 2c,d). TEM images showed particles with sizes of 71.61 ± 24.21 nm (*n* = 10), similar to the sizes of P22-based nanoreactors reported by Chauhan et al. [31]. VLPs sizes might vary due to different amounts of the encapsidated enzyme in each VLP. TEM images of ASNase-P22 nanoreactors also show a compact packaging of ASNase enzyme inside, suggesting a high confinement molarity. In addition, TEM images showed mostly stuffed VLPs (blue arrow) in comparison to empty VLPs (yellow arrow). Additional TEM images are provided in Appendix A.

### 3.2. Characterization of ASNase-P22 Nanoreactors

#### 3.2.1. Composition

Considering the internal volume of the P22 capsid and the volume of the folded ASNase-SP, using Equation (1) results in 26% of occupancy, representing a similar result to that reported by Patterson et al. [41].

To estimate the protein composition of ASNase-P22 nanoreactors, a densitometric analysis of SDS-PAGE of different samples was performed (Appendix A), as previously reported [40]. Our results showed that ASNase-P22 nanoreactors were composed of nearly 60% CP (58.4 ± 7.9%) and 40% ASNase-SP tetramers (41.6 ± 8.1%). Using Equation (2), it was estimated that each nanoreactor was composed of 420 CP and nearly 110 encapsulated ASNase-SP tetramers, resulting in a confinement molarity of 6.58 mM. This result is close to the maximum confinement molarity of 7.16 mM reported by Patterson et al. [41]. Table 2 shows a comparison between different enzymes encapsidated in P22-based VLPs.

#### 3.2.2. ASNase-P22 Nanoreactors Kinetics

The kinetic parameters of ASNase-P22 nanoreactors were determined by varying the Asn concentration from 0 to 5 mM. Results are shown in Figure 3.

The initial rate (V0) of ammonia production increased when substrate concentration reached 1 mM at the enzyme saturation point (Figure 3a). Data were fit to the Michaelis-Menten’s model (Figure 3b). Notice that data showed the typical hyperbola-shaped curve. The obtained Km and Kcat values of ASNase-P22 were 15- and 2-fold higher than free ASNase from *E. coli*, respectively (Table 3). Kcat/Km was one magnitude order lower. These results show that ASNase-P22 nanoreactors have lower affinity and enzymatic efficiency compared to the free enzyme. Decreased enzyme efficiency has also been observed in confined enzymes by other authors. These results could be explained by the restricted conformational changes during catalysis, a lower substrate diffusion rate inside the VLPs, or a non-observable population of encapsidated inactive enzyme [41,45,46].

#### 3.2.3. Stability of ASNase-P22 Nanoreactors

Taking into account that protein stability is a limiting factor in the development of successful biopharmaceuticals based on enzymes, the stability of the ASNase-P22 nanoreactors was studied by two complementary approaches: (A) thermostability, and (B) protein aggregation. Results were compared to the commercial ASNase from *E. coli*, Leunase^®^.

#### 3.2.4. Thermostability

Enzymatic activity of ASNase-P22 nanoreactors was studied at 4, 25, and 37 °C, 4 and 25 °C being the most common storage temperatures and 37 °C, the physiological temperature where enzyme should be active. Results were compared to the commercial formulation of ASNase from *E. coli* (Leunase^®^). The effect of glycerol at 10% *v*/*v* as a nanoreactor-stabilizer was also studied. Figure 4 shows the initial rates of the enzymatic kinetic profile.

At 4 °C, the initial rates of Leunase^®^ and ASNase-P22 nanoreactors, with or without glycerol for up to 168 h of incubation, were similar (Figure 4a–c). At 25 °C, Leunase^®^ and ASNase-P22 nanoreactors, with and without glycerol, showed no difference after 24 h, but at 60 h of incubation, ASNase-P22 nanoreactors decreased 75% of the initial rate, independently of the presence of glycerol (Figure 4d–f). At 37 °C and 12 h of incubation (Figure 4h), the initial rate of ASNase-P22 nanoreactors decreased compared to Leunase^®^ (Figure 4g). After 24 h of incubation, the initial rate was 2-fold decreased (Figure 4h). In contrast, ASNase-P22 nanoreactors with 10% of glycerol showed a similar enzymatic profile compared to the control (Figure 4i) until 24 h of incubation. But at incubation times over 48 h, the reaction efficiency of the ASNase-P22 nanoreactors in the absence or presence of 10% of glycerol decreased by 50% and 25%, respectively, compared to Leunase^®^ (Figure 4g–i).

In summary, our results show that glycerol stabilizes ASNase-P22 nanoreactors for up to 24 h at 37 °C, which represents almost half of the Leunase^®^ lifetime. Other stabilizers are currently under study to improve the thermal stability of ASNase-P22 nanoreactors such as sucrose, trehalose, glucose, maltose, ribose, PEG mixed solvents, pyridinium, and imidazolium-based ionic liquids, as has been proposed [47,48,49].

#### 3.2.5. Protein Aggregation

Proteins have a natural propensity to aggregate due to a combination of Van der Waals forces, hydrogen bonds, disulfide linkages, and hydrophobic interactions. Protein aggregation is a major problem facing the enzyme-based biopharmaceuticals and is a common consequence of environmental factors such as temperature, pH, and ionic force variations, among others [50].

A DLS analysis was performed to detect aggregates in the ASNase-P22 nanoreactors, which were incubated at different temperatures for 12 days in PBS pH 7. Z-averages and distribution of particle sizes are shown in Figure 5. At 4 °C, Leunase^®^ showed 12.9 nm of particle size on average (Figure 5a). ASNase-P22 nanoreactores showed a maximum peak at 71.5 nm (Figure 5b), which corresponds to the expected size of nanoreactors, according to TEM images (Figure 2c,d). However, some aggregates with a diameter of 257.3 nm were also detected. ASNase-P22 nanoreactors, stabilized with 10% of glycerol, showed a peak at 70.7 nm, but higher nanoparticles of 625.5 nm and 5486 nm of diameter were also detected (Figure 5c). At 25 °C, Leunase^®^ maintained a similar distribution of particle sizes compared to 4 °C (Figure 5d). In contrast, ASNase-P22 nanoreactors showed aggregates of 383 nm and 5590 nm diameter (Figure 5e). Surprisingly, ASNase-P22 nanoreactors with glycerol also showed bigger aggregates, with diameters of 549 nm and 5227 nm (Figure 5f). At 37 °C, Leunase^®^ was 100% aggregated, and the sample showed a particle size of 877 nm on average (Figure 5g). ASNase-P22 nanoreactors showed aggregates with a diameter of 1041 nm and 5013 nm, but 40.1% of the non-aggregated sample maintained a size of 86 nm on average (Figure 5h), which corresponds to half of the non-aggregated population at 4 °C or 25 °C (Figure 5b,e, respectively). ASNase-P22 nanoreactors with glycerol showed particles with diameters of 94.9 nm, 675 nm, and 5190 nm (Figure 5i). To clarify the presence of aggregates, we performed DLS analysis of glycerol alone. The bigger peaks in the samples of ASNase-P22 nanoreactors with glycerol seem to be solely due to the glycerol, as is shown in the negative control in Figure 5j (with a maximum at 467 nm and 5039 nm).

These results showed that Leunase^®^ is completely aggregated after 12 days of incubation at 37 °C, in contrast to ASNase-P22 nanoreactors, which showed only 60% of aggregation under the same conditions. No temperature-dependent disassembly of the ASNase-P22 nanoreactors was detected. These observations were confirmed by TEM (not shown).

### 3.3. ASNase-P22 Nanoreactors PEGylation

Immunogenicity of ASNase limits its use as a therapeutic agent, usually due to short lifetimes when exposed to proteases and an elevated immune system response [51]. PEGylation is one of the most common chemical modification of therapeutical proteins, used to improve their stability and to diminish immunogenicity. PEGylation is defined as the covalent attachment of PEG molecules to the surface of proteins. In this work, we were interested in evaluating the stability and immunogenicity of PEGylated ASNase-P22 nanoreactors.

Results of the PEGylation reaction are shown in Figure 6. A scheme of the ASNase-P22 nanoreactors is shown in Figure 6a. The followed PEGylation strategy was based on the NHS/EDC chemistry reported by [52]. First, the carboxylic acid groups of the modified PEG (molecule A) were activated with EDC (molecule B) and NHS (molecule C) for 1 h at 25 °C. Then, an NHS ester was formed and reacted with the primary aminoacids of the P22 capsids, forming a stable amide bond. As can be observed in Figure 6b, the zeta-potential of non-PEGylated ASNase-P22 nanoreactors (−9 mV) was significantly different to PEGylated ASNase-P22 nanoreactors (−14 mV), indicating that electrical charge changed after reaction and, indirectly, confirmed successful PEGylation. PEGylation increases the Z potential of the ASNase-P22 nanorectors, decreasing the chance of a collision. In other words, PEGylation favors the nanoreactors dispersion. Afterward, DLS analysis was performed to evaluate the size of the PEGylated ASNase-P22 nanoreactors. As can be observed in Figure 6c,d, our control-PEG and activated PEG showed a particle size of 4 nm to 5 nm in diameter with a Z-average of 586.0 nm and 502.2 nm, respectively. PEGylated ASNase-P22 nanoreactors showed a particle size of 69.4 nm on average, which is very similar to the non-PEGylated ASNase-P22 nanoreactors of 71.61 ± 24.21 nm obtained by TEM (Figure 2c,d) and DLS (Figure 5b). These results can be explained by the following model: extended PEG5000 molecules feature a length of 27 nm, but considering its conformational flexibility, the size differences of PEGylated and non-PEGylated ASNase-P22 nanoreactor should not be significant [53]. Our results showed that PEGylation of ASNase-P22 nanoreactors was successful, resulting in a significant Z-potential change compared to non-PEGylated VLPs.

### 3.4. PEGylated ASNase-P22 Nanoreactors Stability and Cytotoxicity

In patients with ALL, ASNase is often administrated by intravenous infusion, where it is exposed to proteases that are necessary for hemostasis, fibrinolysis, and tissue repair [54]. ASNase is mostly inactivated by proteases such as cathepsin B and asparagine endopeptidase, expressed by leukemia cells and macrophages [55,56,57,58]. Therefore, the stability of non-PEGylated and PEGylated ASNase-P22 nanoreactors were evaluated in the presence of HBS at 37 °C and different times of incubation (Figure 7). Figure 7a shows that, in the absence of HBS, Leunase^®^ was stable for 24 h, but after 48 h its catalytic activity decreased by 75%, similar to ASNase-P22 nanoreactors (Figure 7b). PEGylated ASNase-P22 nanoreactors showed a decreased reaction rate at 24 h of incubation, probably due to a substrate diffusion impediment by the attached PEG chains to the VLPs surface. After 48 h, the nanoreactor was inactive (Figure 7c). In the presence of HBS, Leunase^®^, non-PEGylated, and PEGylated ASNase-P22 nanoreactors showed no difference compared to the assays with no added HBS (Figure 7d, 7e, and 7f, respectively).

In summary, no ASNase inactivation is attributed to HBS in the kinetic profiles of Leunase^®^ or PEGylated and non-PEGylated ASNase-P22 nanoreactors.

Furthermore, leukemic cell line MOLT-4 (ASNase sensitive T lymphocytes) were exposed to ASNase-P22 nanoreactors at different concentrations to evaluate their cytotoxicity after 72 h of exposure. Results are shown in Figure 8.

Our results show that Leunase^®^, non-PEGylated, and PEGylated ASNase-P22 nanoreactors caused cytotoxicity in MOLT-4 in a concentration-dependent manner. (Figure 8a–c, respectively). However, ASNase nanoreactors showed about 3-fold less cytotoxicity compared to Leunase^®^, probably due to their lower stability at 37 °C (Figure 4g–i), or a lower substrate diffusion into nanoreactors.

In summary, our results show that ASNase-P22 nanoreactors are cytotoxic against leukemic cells; however, further research should be performed to extend their lifetime to be competitive with commercial formulations.

## 4. Conclusions

In this study, ASNase from *E. coli* was encapsidated in bacteriophage P22-based VLPs by a genetically directed strategy. We obtained one of the best molarity confinements reported until now. The encapsidation product was named ASNase-P22 nanoreactor and was fully characterized. Although the enzymatic efficiency and thermostability of ASNase-P22 decreased in comparison to the commercial formulation, Leunase^®^, nanoreactors were functional under physiological conditions such as 37 °C and the presence of human blood serum (HBS) for up to 24 h, when stabilized with 10% glycerol (*v*/*v*). Nanoreactors were coated with PEG, but this modification did not enhance its stability. On the other hand, ASNase-P22 nanoreactors were active against the leukemic cell lines MOLT-4 in a concentration-dependent manner. To our knowledge, this is the first study where ASNase is encapsidated inside of VLPs. We expect that our findings motivate future research focused on ASNase encapsidation inside VLPs to obtain a novel biobetter for ALL treatment with enhanced stability and reduced immunogenicity. Further research will be focused on improving the nanoreactor’s lifetime under physiological conditions and exploring the immunogenicity of non-PEGylated and PEGylated ASNase-P22 nanoreactors in a murine model.

## 5. Patents

This work is deposited on patent application MX/a/2019/012106.

## Figures and Tables

**Figure 1 pharmaceutics-13-00604-f001:**
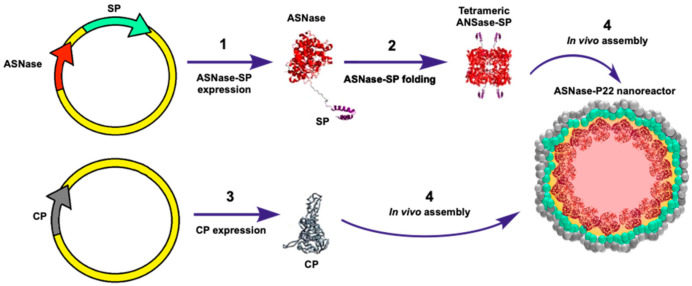
ASNase-P22 nanoreactors. Genetically directed ASNase encapsidation in P22-based virus-like particles (VLPs): (**1**) Cloning and ASNase-SP chimeric protein expression. (**2**) Folding of tetrameric enzyme; (**3**) Capsid protein expression; and (**4**) VLP assembly. ASNase: Asparaginase; SP: Scaffold protein; CP: Capsid protein.

**Figure 2 pharmaceutics-13-00604-f002:**
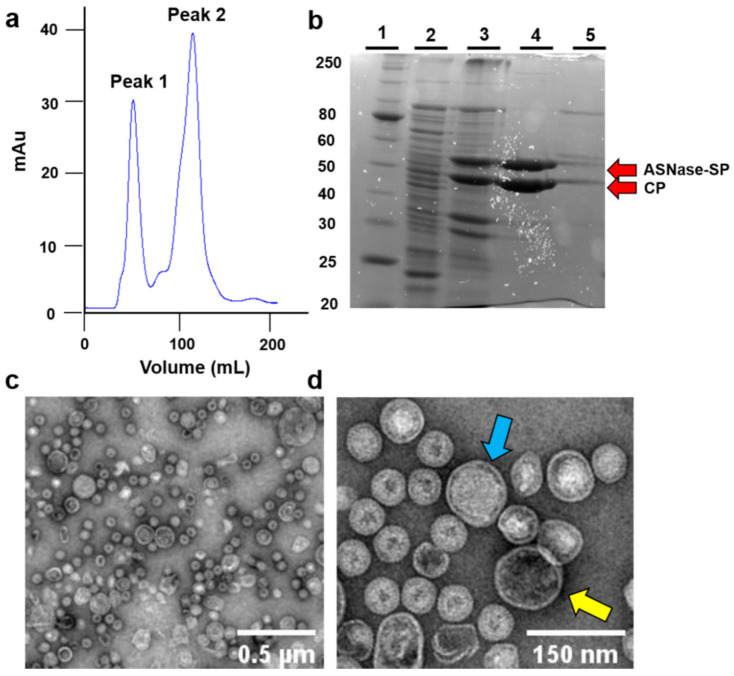
Purification and characterization of ASNase-P22 nanoreactors. (**a**) Size exclusion chromatogram of the ASNase-P22 purification. Peak 1 shows the eluted fraction where the assembled nanoreactors are found. (**b**) SDS-PAGE of the purification steps of ASNases-P22 nanoreactors. Lane 1, molecular weight marker. Lane 2, non-induced cells lysate. Lane 3, induced cell lysate. Lane 4, eluted fraction corresponding to peak 1 of the chromatogram. Lane 5, eluted fraction corresponding to peak 2 of the chromatogram. Red arrows point out the ASNase-SP and CP bands at 53 kDa and 47 kDa, respectively. (**c**,**d**) TEM images of ASNase-P22 nanoreactors found in the eluted fraction corresponding to peak 1 of the chromatogram. Blue and yellow arrows point out a stuffed VLP and an empty VLP, respectively.

**Figure 3 pharmaceutics-13-00604-f003:**
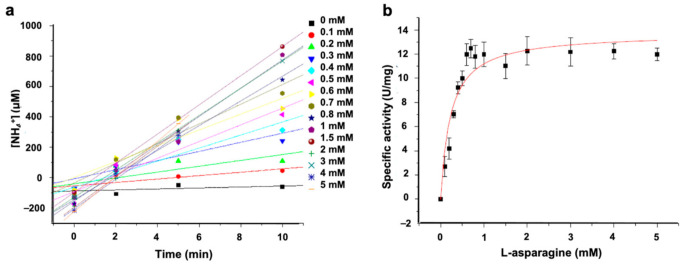
Kinetic parameters of ASNase-P22 nanoreactors based on the Michalis-Menten’s model. (**a**) Initial rates (V0) calculated for different concentration of Asn are indicated on the right. At 5 mM of substrate, the enzyme is saturated. Color codes of the substrate concentration are shown. (**b**) Specific activity fit to the Michaelis-Menten’s model. Chi^2^ = 1.68. R^2^ = 0.84. (*n* = 3). Error bars represent the standard deviation.

**Figure 4 pharmaceutics-13-00604-f004:**
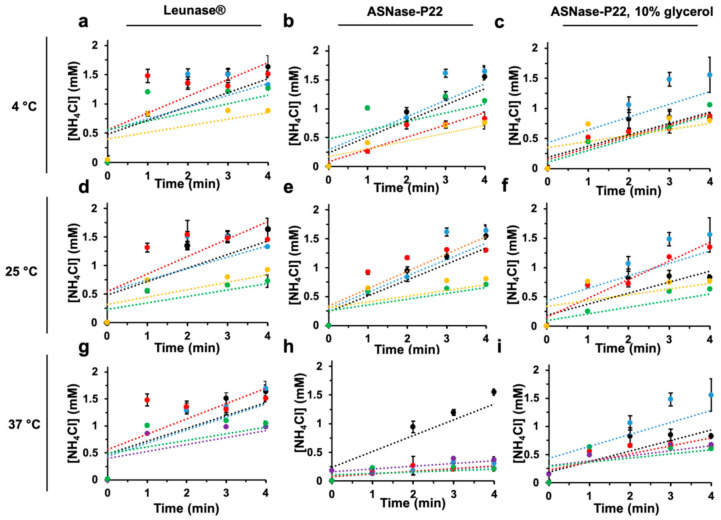
ASNase-P22 nanoreactors enzymatic activity at different times and temperatures of incubation. The initial rates (V0), calculated at 4, 25, and 37 °C are shown, indicated on the left. (**a**,**d**,**g**) Leunase^®^. (**b**,**e**,**h**) ASNase-P22 nanoreactors. (**c**,**f**,**i**) ASNase-P22 nanoreactors with 10% of glycerol. Colored lines represent incubation times of samples at given temperatures, as follows: black, 0 h; blue, 12 h; red, 24 h; purple, 48 h; green 60 h, and yellow, 168 h. (*n* = 3). Error bars indicate standard deviation.

**Figure 5 pharmaceutics-13-00604-f005:**
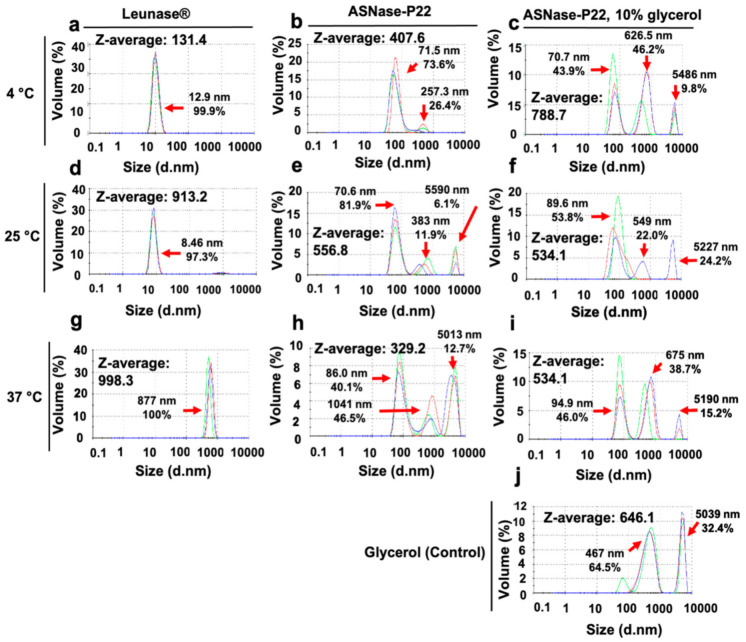
Dynamic light scattering (DLS) profile of ASNase-P22 nanoreactors incubated at 4, 25, and 37 °C. (**a**,**d**,**g**) Leunase^®^. (**b,e,h**) ASNase-P22, and (**c**,**f**,**i**) ASNase-P22 with 10% glycerol. (**j**) 10% glycerol in 50 mM PBS. Representative DLS profiles of at least *n* = 3 are shown. Z-averages, particle sizes (nm), and percentage of volume (mL) for each case are indicated in the figure.

**Figure 6 pharmaceutics-13-00604-f006:**
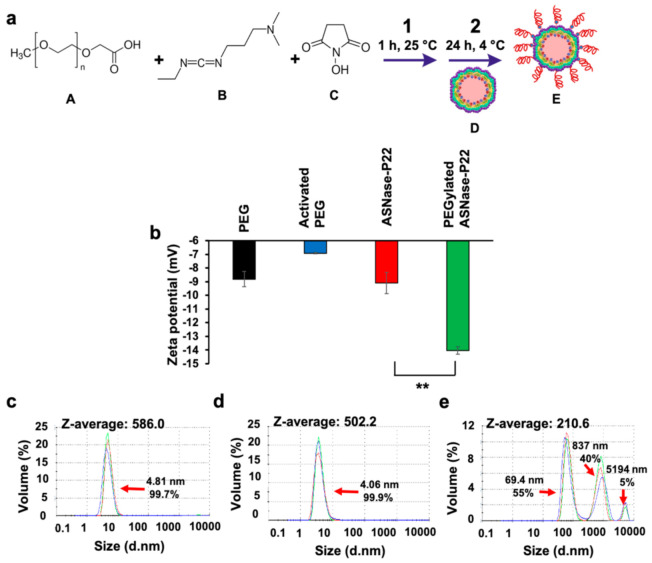
PEGylation of ASNase-P22 nanoreactors. (**a**) PEGylation reaction of ASNase-P22 nanoreactors. Molecule A: PEG carboxylic acid; B: 1-Ethyl-3-(3-dimethylaminopropyl)carbodiimide (EDC); C: N-hydroxysuccinimide (NHS); D: P22-based VLP; E: PEGylated ASNase-P22 nanoreactor. (**b**) Zeta-potential measurements of PEG, activated PEG (as negative controls), non-PEGylated and PEGylated ASNase-P22 nanoreactors. Error bars represent the standard deviation. (**) *p* < 0.01. Size particle measurements of (**c**) PEG and (**d**) activated PEG (as negative controls) and (**e**) PEGylated ASNase-P22 nanoreactors. The figure shows a representative DLS profile of at least *n* = 3. Z-averages, particle sizes (nm), and percentage of volume (mL) for each case are indicated in the figure.

**Figure 7 pharmaceutics-13-00604-f007:**
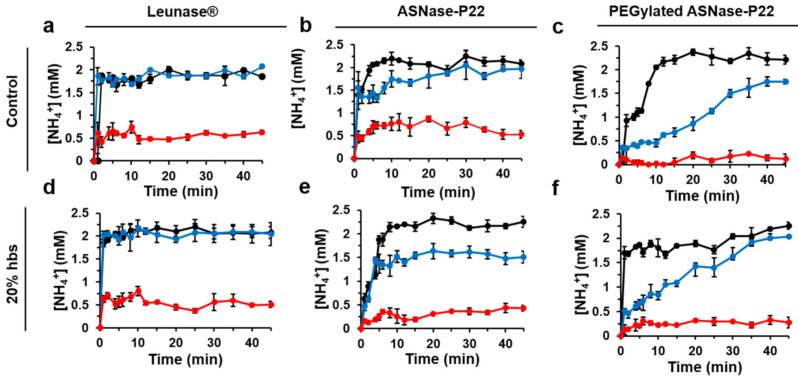
Enzymatic activity of ASNase-P22 nanoreactors in the presence or absence of human blood serum (HBS) at different incubation time at 37 °C. Control conditions without HBS. (**a**) Leunase^®^, (**b**) ASNase-P22, and (**c**) PEGylated ASNase-P22. (**d**) Leunase^®^ +20% HBS, (**e**) ASNase-P22 +20% HBS, and (**f**) PEGylated ASNase-P22 +20% HBS. ASNase-P22 nanoreactors contained 10% glycerol. Black, 0 h; blue, 24 h; red, 48 h. (*n* = 3). Error bars represent the standard deviation.

**Figure 8 pharmaceutics-13-00604-f008:**
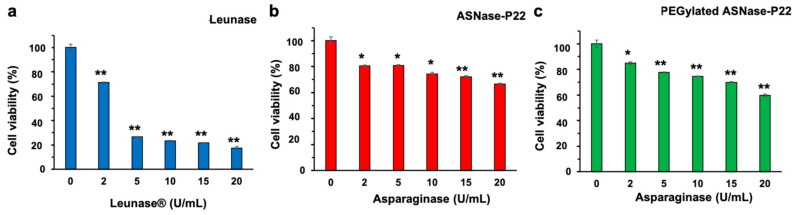
Concentration-dependent cytotoxicity of ASNase-P22 nanoreactors against MOLT-4 cells. (**a**) Control (+) Leunase^®^; (**b**) non-PEGylated; and (**c**) PEGylated ASNase-P22 nanoreactors. (*n* = 3). Error bars represent the standard deviation. (*) *p* < 0.05, (**) *p* < 0.01.

**Table 1 pharmaceutics-13-00604-t001:** Primers used for *asnB-sp* construction.

Encoded Protein	Primer	Sequence	Restriction Site
*E. coli asnB* ASNase II	*asnB* Fw	GATATACCATGGCATTACCCAATATCACC	NcoI
	*asnB* Rv	CCGGCTCGAGGTACTGATTGAAGATCTGCT	XhoI
P22 Scaffold protein (*sp*)	*sp* Fw	ATATCTCGAGCTGGTGCCGCGCGGCAG	XhoI
	*sp* Rv	TCTCGAATTCTTATCGGATTCCTTTAAG	EcoRI

**Table 2 pharmaceutics-13-00604-t002:** Enzymes encapsidated in bacteriophage P22-based VLPs.

Enzyme	Monomers per Capsid	Internal Radium of Capsid (nm)	M_conf_	Occupancy (%)	Reference
CelB glycosidase	87	24	NR	NR	[44]
CYP450	110	22	3.14	NR	[37]
ASNase	448	30	6.58	26	This work
Alcohol dehydrogenase	249	24	7.16	27	[41]

NR: Not reported.

**Table 3 pharmaceutics-13-00604-t003:** Kinetic parameters of ASNase-P22 nanoreactors compared to the native ASNase of *E. coli*.

Sample	K_cat_ (s^−1^)	K_m_ (mM)	K_cat_/K_m_ (M^−1^ s^−1^)	Specific Activity (U/mg)	V_max_ (µM/min)	Reference
ASNase-P22	49	0.227	2.19 × 10^5^	13.73	131.78	This work.
ASNase (*E. coli*)	24	0.015	1.60 × 10^6^	NR	NR	[59]

## Data Availability

Data is contained within the article or Appendix A.

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
