# Peer review of "Asparaginase-Phage P22 Nanoreactors: Toward a Biobetter Development for Acute Lymphoblastic Leukemia Treatment"

_pharmaceutics, 2021, doi:10.3390/pharmaceutics13050604_

Round 1

Reviewer 1 Report

  1. Figure 3, legend: color "code" are shown - change to "codes" or "are" to "is".
  2.   Figure 3, b) not clearly visible the unit of measurement (U/?)on the Y-axis. 
  3. Mat.& Methods, line 103: MgCl2 change to MgCl2.
  4. Results, line 239: it is said that "data not shown". However, in Figure 2, track 3 these data are presented. 
  5. Results, line 272: "TEM images of the ASNase-P22 nanoreactors suggest a high rate of ASNase encapsidation":  is it really visible by TEM analysis? 
  6. Results, line 277-279: "...a densitometric analysis was performed...Our results showed that ASNase-P22 nanoreactors were composed of nearly 50% CP (48.4±7.9%) and 50% ASNase tetramers (51.6±8.1%), resulting in a ratio of 1:1.07 (CP: ASNase)". Questions: a) is there thought the densitometric analysis of the gel?, b) probably should be written ASNase-SP, not the ASNase? In Figure 3, track 4 the zone for ASNnase looks bigger. 
  7. Results, line 339-340: there are two times "ASNase-P22" in the sentence "the reaction efficiency of the ASNase-P22 and ASNase-P22 nanoreactors". Also "...decreased 50%
    and 25%, respectively, compared to Leunase® (Figure 4g-i)" - is there thought the decrease by or to ...%?
  8. Results, lines 375-376: Not very clear "In summary, our results show that glycerol stabilizes ASNase-P22 nanoreactors for up to 24 h at 37° C which represents nearly half of the time, compared to Leunase®".
  9. Results, line 384: " ...many environmental factors such as temperature". maybe better without "many" and as temperature, ... and ... 
  10. Figure 6, line 487: there is a mistake instead of the second D should be E. 
  11. General suggestion:

authors note that the use of ASNase is restricted partly by its immunogenicity. However the same be true with the use of proposed nanoreactors. Phage-like particles can be very immunogenic,

Reviewer 2 Report

The authors describe an alternative strategy to improve the pharmacodynamic and pharmacokinetic of L-Asparaginase using  Virus-like particles.

  • Why do the authors think this strategy is more effective than for instance using liposomes? No reduction in terms of enzymatic activity was observed after loading this enzyme in liposomes. Moreover, an increase on the half-life of the enzyme following intravenous injection was also achieved.
  • Which will be the route for administering the developed Asparaginase system?
  • What do the authors think about the very high mean size of developed system in terms of in vivo biodistribution profile?
  • In table 2 the authors indicate the % of L-Asparaginase occupancy in the bacteriophage P22-based VLPs. Could the authors translate this parameter in terms of amount of enzyme and the respective enzymatic activity (U/mg) ?
  • In Figure 6 B the authors show the zeta potential of PEG, activated PEG (as negative controls), non-PEGylated and PEGylated ASNase-P22 nanoreactors. How do you explain that ASNase-P22 nanoreactors are more negatively charged?
  • In the legend of Figure 8 the authors have written “Concentration-dependent cytotoxicity of ASNase-P22 nanoreactors against MOLT-4 cells.”. However in figures 8a,b and c it is represented the absorbance of cells upon incubation with leunase, ASNase-P22 and PEGylated ASNase-P22. The representation should be in terms of cellular viability considering that formulations tested with 0 U/ml correspond to 100% viability. Do the authors think that ASNase-P22 nanoreactors are cytotoxic towards leukemic cells?

Reviewer 3 Report

The manuscript of Cristina Díaz-Barriga et al. seems to be the first approach where ASNase is encapsulated inside of a virus like particles. Ussually ASNase is trapped inside of a erythocyte.  The introduction is clearly written and provides necessary background. Experiments are designed and performed correclty. Although I do have a small concern. I would like to see the TEM pictures, because it seems that the authors have some aggregation problem. I  am not sure if it has an application potential if the concruct tends to aggregate. More studies in this direction is needed (e.g. TEM).

Round 2

Reviewer 3 Report

All issues has been solved.